# Unveiling health disparities: Diagnostic prevalences in a transgender cohort versus matched controls

Laurel Hiatt[1]*, Blessing S. Ofori-Atta[2], Amanda V. Bakian[3], Nicole L. Mihalopoulos[4], Brooks R. Keeshin[5], Anna Docherty[3], Michael Staley[6], Alison Fraser[7], Emily Sullivan[8], Erin A. Kaufman[3], Hilary Coon[3], Anne V. Kirby[9]

1 Department of Human Genetics, University of Utah, Salt Lake City, Utah, United States of America, 2 Study Design and Biostatistics Center, University of Utah, Salt Lake City, Utah, United States of America, 3 Department of Psychiatry, University of Utah, Salt Lake City, Utah, United States of America, 4 Department of Pediatrics, University of Utah, Salt Lake City, Utah, United States of America, 5 Division of Child Protection and Family Health, University of Utah, Salt Lake City, Utah, United States of America, 6 Office of the Medical Examiner, Utah Department of Human Health and Services, Salt Lake City, Utah, United States of America, 7 Utah Population Database, University of Utah, Salt Lake City, Utah, United States of America, 8 Zero Suicide Program, University of Utah, Salt Lake City, Utah, United States of America, 9 Department of Occupational and Recreational Therapies, University of Utah, Salt Lake City, Utah, United States of America

* laurel.hiatt@hsc.utah.edu

## Abstract

### Importance

Transgender and gender-diverse (TGD) individuals are at risk for discrimination and inequities across legal, social, and medical contexts. Population-level resources have rarely been used for TGD health research and, therefore, data is lacking about prevalences of a wide range of clinical conditions among TGD populations.

### Objective

To leverage the Utah Population Database's demographic, vital, and health records and examine population-level diagnostic prevalences in TGD individuals and an age-matched general cohort.

### Participants

6,664 TGD individuals were identified using ICD codes for gender incongruence between 1995 and 2021; 64,124 age-matched individuals comprised the control cohort.

### Design

Using Phecodes to collapse ICD codes, this study examined differences in the prevalence of medical, mental health, and neurodevelopmental clinical phenotypes in TGD and control cohorts using modified Poisson regression models.

**Data availability statement:** Under the usage agreement with the Utah Population Database (UPDB), we are not permitted to provide publicly available data. UPDB data is not publicly available to protect individual privacy, ensure ethical research practices, and maintain data security. Access is granted on a project-specific basis after thorough review and approval by the Resource for Genetic and Epidemiologic Research (RGE) and IRB. We would also like to note that TGD individuals are especially vulnerable given the current political climate, and the risk of identification warrants additional caution that underscores the necessity of abiding by the policies of UPDB data privacy. Data underlying results are available through the UPDB; research requests can be made at https://uofuhealth.utah.edu/huntsman/utah-population-database/services/research-requests.

**Funding:** The authors receive research funding from the following institutes: National Center for Advancing Translational Sciences (R01MH123489), National Center for Research Resources (R01RR021746), National Center for Research Resources (UM1TR004409), and National Cancer Institute (P30CA2014). Specific funding for this project was received from National Center for Advancing Translational Sciences (UL1TR002538).

**Competing interests:** The authors have declared that no competing interests exist.

## Setting

Affiliated healthcare systems within the state of Utah.

## Main outcome and measure

We evaluated adjusted prevalence ratios of identified Phecodes.

## Results

The TGD cohort showed broadly higher documented prevalences of medical, mental health, and neurodevelopmental conditions compared to controls. Medical diagnoses more common in the TGD cohort included sleep disorders and chronic pain. Disparities in diagnoses such as "other endocrine disorders" and "need for hormone replacement therapy" likely reflect gender-affirming treatments. Mental health conditions including mood, depression, anxiety, and personality disorders were significantly more prevalent in the TGD cohort.

## Conclusions and Relevance

This study highlights diagnostic disparities for TGD individuals across multiple clinical categories. Our findings may be driven by: 1) discrimination and over-medicalization of TGD individuals, 2) differences in accessing and interacting with the healthcare system, and 3) variation in the true incidence of medical and mental health outcomes in the TGD vs control cohorts.

## Background and significance

Transgender and gender-diverse (TGD) populations include individuals whose genders do not match their sex assigned at birth. While the experiences of TGD people are inherently broad—including language- and culture-specific identities—there are well-documented commonalities among individuals within the US and globally [1]. TGD people are at elevated risks for personal and structural inequities relating to legal, social, and medical discrimination compared to cisgender people (those whose assigned sex at birth corresponds to their gender) [2]. These collective and cumulative factors lead to decreased well-being in the TGD population in addition to constructing or augmenting barriers to accessing sufficient healthcare support and services [3]. This results in what has been called the stigma-sickness slope: stigma manifesting as structural discrimination pairs with inadequate or negative healthcare experiences to increase outcomes of sickness and even mortality [4]. There is extensive evidence of these systemic barriers, as well as TGD people's negative encounters with and underutilization of healthcare services. However, there is limited granularity as to the specific nature of the health disparities for which the TGD community is at risk.

Several aspects of TGD health research need to be addressed to bridge these gaps in the literature. Firstly, sampling is a known challenge in TGD research [5–7].

Self-report research is common and maximizes individuals' abilities to self-identify and report on their own experiences, which is a notable strength in research on marginalized groups.(Turban et al. 2023) However, these approaches are also affected by selection biases that may limit generalizability. Self-report studies can comprise probability or non-probability sampling methods, which may identify different patterns of results [7]. For example, while analysis of responses from the Behavioral Risk Factor Surveillance System suggests that the TGD population has a higher prevalence of disability and activity limitations, this finding was not replicated in a
population-based probabilistic sampling study [7]. Secondly, sample size limitations are common in TGD health studies. The TransPOP study, which assessed health and healthcare access for TGD individuals through probabilistic sampling, was limited to 274 TGD participants as a consequence.(Feldman et al. 2021) Typical samples, particularly regarding specific health conditions, are often on the order of hundreds of participants, which lack the necessary resolution to explore within-group analyses [8]. TGD identities are frequently combined into a larger umbrella alongside sexual minority persons [9–11], which further reduces research precision and specificity. A notable exception is the U.S. Transgender Study, a self-report study of 27,715 respondents, which facilitated research on TGD individuals stratified by gender modality [12]. Lastly, TGD research is often restricted to specific clinical categories, such as sexual health or mental health diagnoses; broader awareness of how health inequity may manifest in other diagnostic prevalences is still evolving.

The current study aimed to identify a TGD cohort in a statewide healthcare population and conduct an overview of the prevalence of diagnostic codes used in their medical records. Medical record-based analyses come with their own limitations, such as necessitating that TGD individuals have accessed healthcare, have disclosed their identity to healthcare providers, and that their providers have documented their TGD identity and health needs in ways that can be captured. Nonetheless, they offer the potential to draw valuable conclusions about TGD populations meeting these criteria. While prior U.S. records-based studies have focused on specific medical systems or payer groups [5], the current study accessed data inclusive of multiple systems and payers across the state of Utah.

Using the available ICD record for each cohort member, we examined phenotypes from TGD health literature with clinical and community relevance [13,14]. We aim to address the current lack of population-level prevalence estimates of health conditions among TGD healthcare recipients. We identify an age-matched cohort to contrast diagnostic prevalence rates with the general populace. We leverage Phecodes, collections of ICD codes intended to capture clinically meaningful concepts for research by corresponding to specific disease phenotypes, in order to assess distinct diagnostic umbrellas.(Bastarache 2021) Our approach lays the groundwork for future TGD health research on health needs, health disparities, and interventions to address healthcare needs. It is especially urgent to conduct studies such as this to create an infrastructure for equitable and evidence-based TGD healthcare, as demand is expected to increase dramatically in parallel with an anticipated doubling of the size of the TGD population [15,16].

## Materials and methods

### Data sources

The Utah Population Database (UPDB) securely links the state's demographics, vital records, medical records, and genealogical data [17]. It supports genetic, epidemiological, demographic, and public health research, with vital records from as early as 1904. The UPDB has strict protocols in place to protect data security, and data is analyzed in a secure computing environment. Projects are reviewed and regulated prior to study initiation and annually by the University of Utah Institutional Review Board (IRB) and the Utah Resource for Genetic and Epidemiological Research (RGE) committee.

For this project, UPDB staff queried and linked demographic and vital records with diagnostic data from the Utah Department of Health and Human Services: 1) All Payers Claims Database (reported to the state by insurers), and 2) Inpatient, Ambulatory Surgery, and Emergency records from across the state, and the University of Utah Health system, a major health provider in the state.

### Cohorts

**TGD.** To identify a TGD cohort, UPDB coders queried medical records from the sources above for ICD codes indicating what is referred to clinically as gender incongruence [ICD-10 codes: *F64.0, F64.1, F64.2, F64.8, F64.9, Z87.890*; ICD-9 codes: *302.5, 302.51, 302.52, 302.53, 302.6, 302.85*]. Based on the available data resources, individuals were identified between 1995 and 2021, with robust outpatient data from 2013 onward. Individuals needed to be at least ten years of age at the time of receipt of a gender incongruence diagnostic label and have linked to data in the UPDB; individuals did not need to reside in Utah but had to have received healthcare in the state.

Significant attention was given to case status review by the UPDB team to avoid classification errors; manual review was performed by authors with experience relevant to EHR review. While we likely do not capture every TGD individual who has interacted with systems covered within the UPDB, our data suggests that we have successfully captured a cohort of TGD patients. Additional forms of evidence supporting TGD status were used to supplement the diagnostic data (Figs 1 and 2) [18,19]. The codes used are listed in S3a–S3c Tables.

When feasible, UPDB staff assigned an inferred "gender characteristic" (i.e., transmasculine, transfeminine, nonbinary) to an individual based on procedures, medications, and/or self-report. Self-reports of gender were available from one data source (University of Utah Health); only the most recent self-reported gender was available. Furthermore, our clinical team (NM, BK) gathered and reviewed lists of prescription medication classes and procedure codes used in gender-affirming care. This team then indicated combinations of codes that would indicate

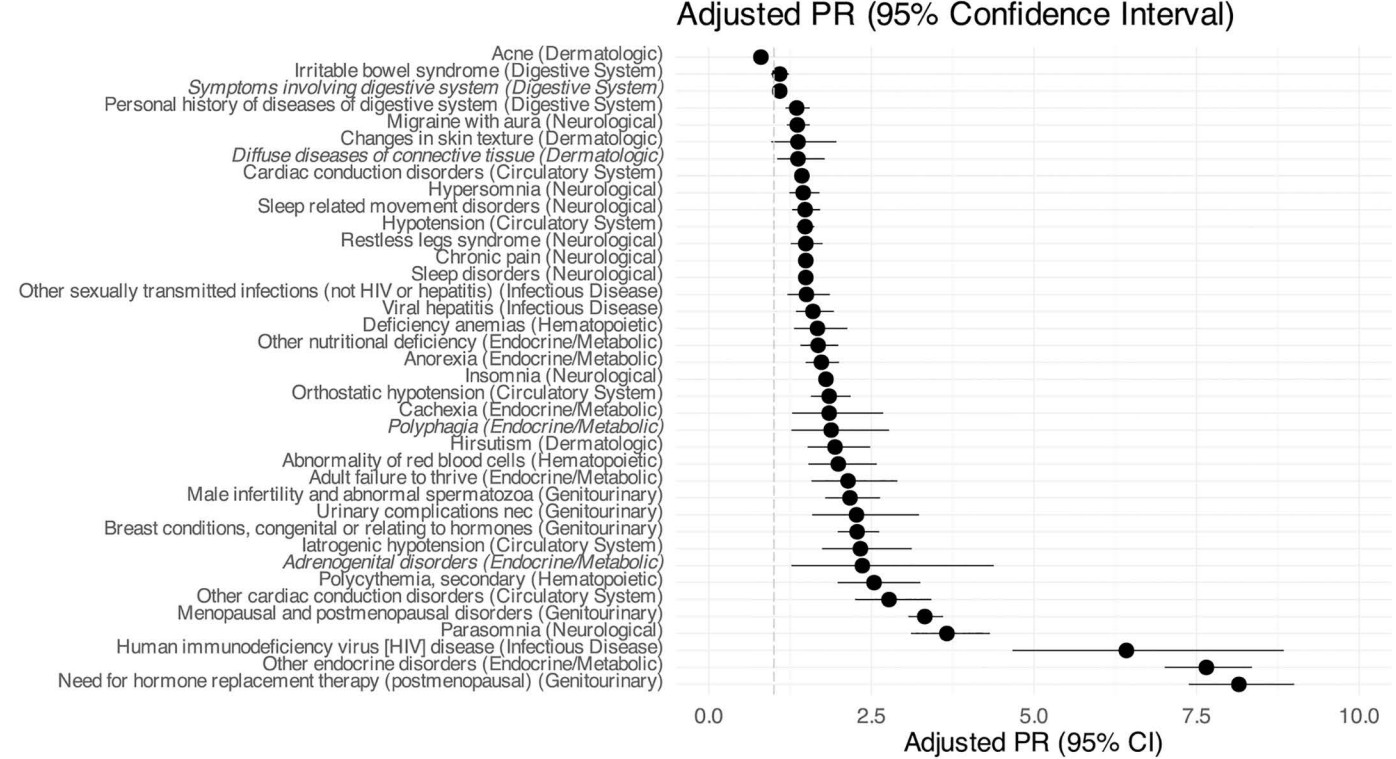

**Fig 1. Forest plot for medical clinical phenotypes.** Depicted is the prevalence ratios (PR) for the transgender cohort compared to the matched cohort, adjusted for adjusted for individual's administrative sex, race, ethnicity, and birth year as well as the log of individual codes as a proxy for healthcare encounters. The dashed gray line indicates 1, where odds are the same between cohorts; ratios greater than 1 indicate a diagnosis is more prevalent in the TGD cohort. Non-significant results are italicized. Detailed information available in S1 Table.

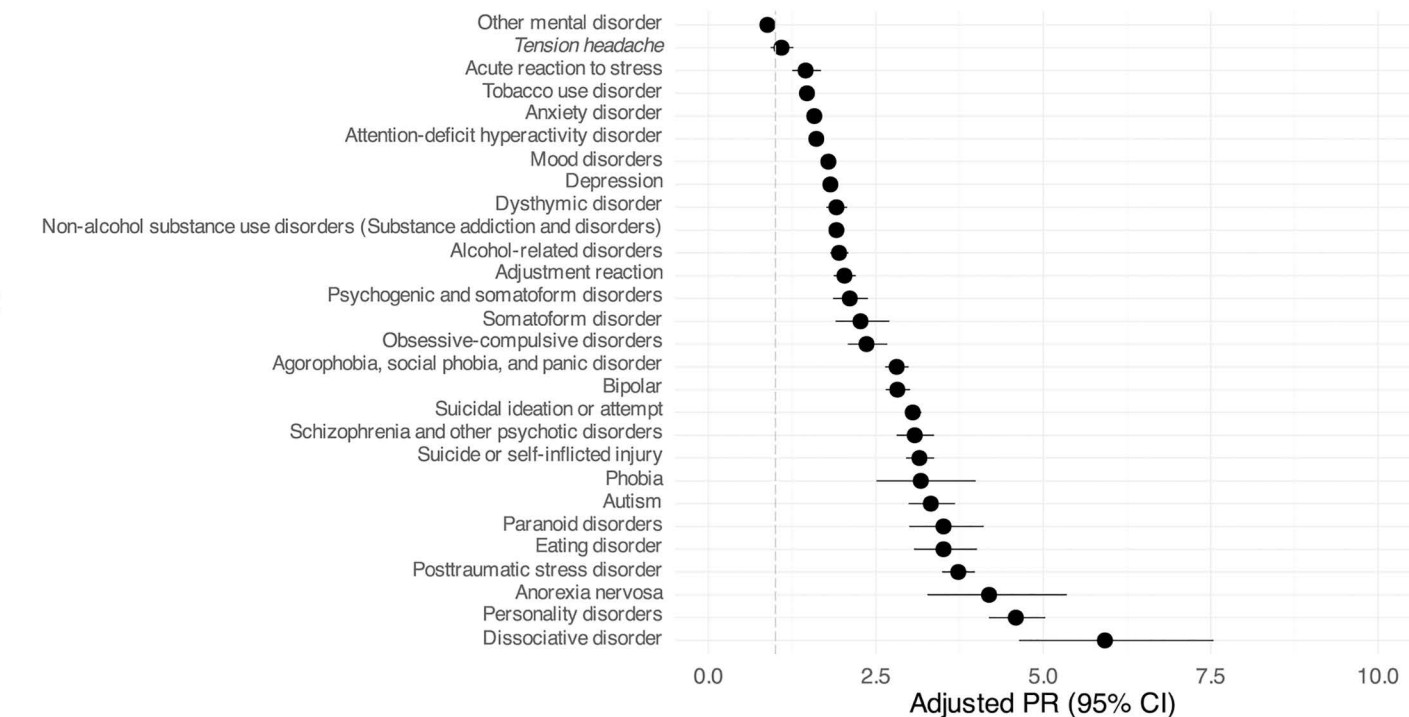

**Fig 2. Forest plot for mental health and neurodevelopmental clinical phenotypes.** Depicted is the prevalence ratios (PR) for the transgender cohort compared to the matched cohort, adjusted for adjusted for individual's administrative sex, race, ethnicity, and birth year as well as the log of individual codes as a proxy for healthcare encounters. The dashed gray line indicates 1, where odds are the same between cohorts; ratios greater than 1 indicate a diagnosis is more prevalent in the TGD cohort. Non-significant results are italicized. Detailed information available in S2 Table.

gender-affirming care, including distinctions based on sex assigned at birth where relevant. When applicable, each prescription and surgery was labeled as masculinizing or feminizing. The UPDB team carefully reviewed cases to assign the appropriate gender characteristic. Cases were flagged for further review when information was inconsistent. Self-reported gender was prioritized unless the self-report date was notably older than other sources of gender characteristic information.

**Control cohort.** A control cohort was selected based on matching criteria and a lack of evidence indicating TGD status. The UPDB team matched ten comparison individuals for each TGD case. Controls were matched on birth decade and number of years living in the state, and each TGD case was matched to five male and five female controls based on sex assigned at birth. We excluded 8,309 individuals from both the Control and TGD cohorts who had ambiguous data signals that indicated they could possibly be TGD (e.g., both male and female sex or gender labels in records from different sources), that could be easily attributed to data entry errors rather than indicative documentation of gender status. The final Control cohort was further reduced by removing individuals who did not have ICD data available in their UPDB record.

## Variables

**Demographic data.** Birth year, birth certificate sex, race, and ethnicity were identified from available linked administrative sources within the UPDB. While there were numerous sex variables linked to the UPDB, for maximal

consistency, we chose to use only birth certificate sex in the analyses. Additionally, we created a 'gender characteristics' variable to reflect the available evidence related to a participant's gender, as described previously [20]. Based on the types of gender-affirming care received, cases were assigned as 'transmasculine' or 'transfeminine' when possible. When self-report data was available from EHR, that was prioritized for categorization (including 'non-binary,' which could only be classified through self-report).

**Diagnoses.** We categorized pertinent ICD codes using the hierarchical Phecode system [21] (S4a–S4b Tables for Phecodes used). We utilized Phecodes (phenotypic codes) to group multiple ICD codes into a collective diagnostic code where appropriate [21]. For Phecode mapping to ICD codes, we followed the guidance outlined in Version 1.2, available on the PheWAS Catalog Website (http://phewascatalog.org) and used the integrated files in the PheWAS package in R [22].

## Analysis

We compared demographic characteristics and Phecode diagnoses (across medical, mental health, and neurodevelopmental phenotypes) between the TGD and Control cohorts using chi-square tests, reporting counts and percentages. Further, we examined demographics and diagnoses by gender characteristics in the TGD cohort (i.e., between transfeminine and transmasculine groups; there were insufficient sample sizes for nonbinary and missing/unknown categories).

We performed univariable and multivariable analyses to evaluate the difference in the prevalence of co-occurring conditions between TGD and Control cohorts, as well between the subgroup of transmasculine versus transfeminine individuals while accounting for the number of conditions per person using modified Poisson regression with robust standard errors [23]. This approach allows us to directly estimate prevalence ratios. Because the Poisson distribution underlies the model, we employed the robust variance estimation method for accurate inferencing. First, unadjusted prevalence ratios were modeled. Next, adjusted prevalence ratios were modeled where the multivariable models included a log of number of conditions, which serves as a proxy for healthcare utilization, as an offset to help control for informative presence bias. The model was adjusted for individual's birth certificate sex, race, ethnicity, and birth year [24]. To address the issue of multiplicity—specifically, multiple endpoints—[25] we applied the Bonferroni correction, controlling the Familywise Error Rate (FWER) and reporting adjusted p-values.

We exponentiated regression coefficients and reported results as prevalence ratios with 95% confidence intervals (CIs). All tests were two-sided, and statistical significance was assessed at a threshold of 0.05 following posthoc Bonferroni correction. Analyses were done using R version 4.3.1 [26].

## Results

Using ICD codes in the UPDB from 1995–2021, we labeled 6664 individuals as TGD and matched them to a final sample of 64124 Control individuals (after removal of 2516 (<4%) controls without ICD data). Of note, estimating a TGD population prevalence was outside the scope of this study. While population prevalence is possible within UPDB, the research team did not have permission to access the relevant denominators for this study.(Smith et al. 2022).

## Demographic

Table 1 contains descriptive data on the TGD and Control cohorts (TGD total cohort descriptive data separated by gender characteristics available in S5a Table). Of the 6664 individuals, 3222 (48%) were classified as transmasculine, 2638 (40%) as transfeminine, and 262 (4%) as nonbinary (Table 1). Specific gender characteristics could not be determined or were inconclusive for 542 (8%) of the cohort: these individuals self-reported as "Other" or "Choose not to disclose" *or* had a qualifying ICD code without further clarifying evidence regarding gender characteristics. Birth certificate sex was

**Table 1. Demographic characteristics.**

| Characteristics | TGD Cases (N = 6664) N (%) | Population Controls (N = 64124) N (%) | p-value |
|---|---|---|---|
| Gender Characteristics | | | |
| Transmasculine | 3222 (48%) | – | – |
| Transfeminine | 2638 (40%) | – | – |
| Non-binary[a] | 262 (4%) | – | – |
| Unknown[b] | 542 (8%) | – | – |
| Administrative Sex (Birth Certificate Sex) | | | <0.001[c] |
| Male | 1325 (20%) | 13684 (21%) | – |
| Female | 1830 (27%) | 15515 (24%) | – |
| Unknown/Missing | 3509 (53%) | 34925 (54%) | – |
| Birth Cohort | | | <0.001[c] |
| 1920s | 17 (0%) | 156 (0%) | – |
| 1930s | 56 (1%) | 523 (1%) | – |
| 1940s | 124 (2%) | 1229 (2%) | – |
| 1950s | 234 (4%) | 2376 (4%) | – |
| 1960s | 299 (5%) | 3111 (5%) | – |
| 1970s | 352 (5%) | 3787 (6%) | – |
| 1980s | 826 (12%) | 8915 (14%) | – |
| 1990s | 2046 (31%) | 22097 (34%) | – |
| 2000s | 1989 (30%) | 21619 (34%) | – |
| 2010s | 29 (0%) | 311 (0%) | – |
| Unknown/Missing | 692 (10%) | 0 (0%) | – |
| Race | | | <0.001c |
| American Indian or Alaska Native | 34 (1%) | 314 (0%) | – |
| Asian | 54 (1%) | 613 (1%) | – |
| Black or African American | 75 (1%) | 772 (1%) | – |
| Native Hawaiian or Other Pacific Islander | 16 (0%) | 345 (1%) | – |
| White | 5185 (78%) | 52083 (81%) | – |
| Multiple Races | 278 (4%) | 2632 (4%) | – |
| Unknown/Missing | 1022 (15%) | 7365 (11%) | – |
| Ethnicity | | | <0.001c |
| Hispanic | 860 (13%) | 10458 (16%) | – |
| Non-Hispanic | 4642 (70%) | 42935 (67%) | – |
| Unknown | 1162 (17%) | 10731 (17%) | – |

Demographic characteristics subset for TGD cases and population controls, with group differences.

[a]Nonbinary could only be identified through self-report; other individuals who identify as nonbinary are likely categorized elsewhere.

[b]Unknown includes individuals who self-reported "Other" (n = 335) and "Choose not to disclose" (n = 1), as well as individuals who had a qualifying ICD code but no other data that could be linked with gender characteristics.

[c]Chi-squared test.

missing or unknown for 3509 (53%) of TGD individuals and 34925 (54%) of the control cohort, indicating consistency across cohorts. The missing or unknown across both groups strongly reflects individuals born out of state, which was incorporated into the matching process (time living in the state). Birth cohorts (determined by decade of birth) span from the 1920s to the 2010s, with the highest frequencies of cohort members born in the 1990s and 2000s.

Once accounting for "Unknown/Missing" values in race and ethnicity—1022 (15%) and 1162 (17%), respectively–our aggregate cohort was broadly comparable (defined as within a percentage point) to 2023 Utah Census data [27]. Hispanic ethnicity in our data was within 0.5% of the Census report's estimate of Hispanic/Latino identity. Differences in race proportions compared to the Census were limited to Asian Americans (underrepresented in the TGD cohort by 2%) and multiracial individuals (overrepresented by 1.3%). When comparing the cohorts, the TGD cohort has significantly more American Indian/Alaska Native individuals (1% versus 0% Control, p-value < 0.001) and fewer Hispanic/Latino individuals (13% versus 16% Control, p-value < 0.001).

## Clinical Results

Fig 1 (S1 Table) details the prevalence of medical clinical phenotypes previously identified in the literature and community spaces as relevant to TGD health across eight categories identified in phenome-wide association studies (PheWAS). These categories are endocrine/metabolic, infectious disease, circulatory system, dermatologic, digestive system, hematopoietic, and neurological. As the TGD cohort showed increased prevalence across all PheCodes, even when controlled for the number of codes as a proxy for healthcare engagement, we elected to focus on these subset phenotypes in our primary analyses. The most frequent medical diagnoses in the TGD cohort are *sleep disorders* (TGD: 35%, GEN: 15%; aPR 1.49), *insomnia* (TGD: 27%, GEN: 10%; aPR 1.80), *other endocrine disorders* (TGD: 17%, GEN: 1%; aPR 7.65), and *chronic pain* (TGD: 17%, GEN: 7%; aPR 1.49). The prevalence ratio (PR) comparison between the TGD and control cohorts is shown, as is a PR adjusted (aPR) for birth cohort, birth certificate sex, race, and ethnicity, with the number of unique ICD codes as an offset. Broadly, the TGD cohort showed increased PRs compared to the control cohort across all categories. The diagnoses with the highest aPRs are: *need for hormone replacement therapy (postmenopausal)* (aPR 8.15, 95% CI 7.38–9.00, p-value <0.001), *other endocrine disorders* (aPR 7.65, 95% CI 7.01–8.35, p-value <0.001), and *human immunodeficiency virus (HIV) disease* (aPR 6.42, 95% CI 4.67–8.84, p-value <0.001).

Fig 2B (S2 Table) reviews the prevalence ratios of mental health and neurodevelopmental clinical phenotypes. The most common mental health/neurodevelopmental diagnoses in the TGD cohort are *mood disorders* (TGD: 70%, GEN: 25%; aPR 1.79), *depression* (TGD: 68%, GEN: 24%; aPR 1.82), *anxiety disorder* (TGD: 64%, GEN: 26%; aPR 1.58), and *other mental disorder* (TGD: 53%, GEN: 40%, aPR 0.88). The mental health/neurodevelopmental diagnoses with the highest aPRs between cohorts are: *dissociative disorder* (aPR 5.92, 95% CI 4.64-7.54, p-value <0.001), *personality disorder* (aPR 4.59, 95% CI 4.19-5.03, p-value <0.001), *anorexia nervosa* (aPR 4.19, 95% CI 3.27-5.35, p-value <0.001), and *posttraumatic stress disorder* (aPR 3.73, 95% CI 3.49-3.98, p-value <0.001).

A subset of clinical phenotypes showed statistically significant differences based on gender characteristics (S5b–S5c Tables). These include clinical diagnoses such as *viral hepatitis* (TF: 3%, TM: 1%, aPR 0.44, 95% CI 0.28–0.70, p-value = 0.02), *HIV disease* (TF: 2%, TM: < 0.3%, aPR 0.18, 95% CI 0.09–0.40, p-value <0.001), *attention-deficit hyperactivity disorder* (ADHD) (TF: 21%, TM: 19%, aPR 0.73, 95% CI 0.63–0.84, < 0.001), and *autism* (TF: 9%; TM: 6%, aPR 0.48, 95% CI 0.37–0.63, < 0.001). These also include phenotypes expected to be influenced by phenotypic sex, such as: *need for hormone replacement therapy (postmenopausal)* (TF: 18%, TM: 12%, aPR 0.73, 95% CI 0.62–0.86, p-value = 0.008), and *male infertility/abnormal spermatozoa* (TF: 5%, TM: < 0.3%; aPR 0.07, 95% CI 0.02–0.17, p-value <0.001). Some of the statistically different prevalences between gender characteristics can be explained by specific hormone therapy, such as *secondary polycythemia* (TF: 1%, TM: 2%, aPR 3.20, 95% CI 1.67–6.13, p-value = 0.015), which can be induced by testosterone.

## Discussion

We identified a TGD cohort of 6664 individuals who received healthcare in the state of Utah over an approximately 26-year period. This cohort ascertainment was made possible through the unique advantages of the UPDB. We anticipate that the identified cohort is likely the lower bound of TGD individuals in the capture area, as individuals who have not sought gender-affirming care, who have received gender-affirming care elsewhere, who have healthcare access barriers,

or who have largely avoided care due to concerns of discrimination may not be identified based on our ascertainment methods [28,29]. Our data supports previous research suggesting that American youth and younger generations are more likely to identify as transgender, possibly due to increased awareness and social support [30]. That being said, our data resources are also more calibrated to capture the lifetime healthcare utilization of younger patients compared to older individuals who may have clinical records outside of what is documented electronically by the UPDB.

## Medical phenotypes

We present diagnostic prevalences across health categories in a TGD cohort, with direct comparison to an age-matched control cohort. Much of our data supports previous observations pertaining to TGD health, including sexual and mental health comorbidities [31]. For example, our findings of elevated prevalence of dietary/nutrition-related phenotypes—including anorexia prevalence of 3% compared to 1% in the control cohort and 0.6% in the general population— [32] follows established scholarship regarding eating disorder prevalence in TGD communities [33]. Notably, we found these phenotypes (as well as the eating disorder and anorexia nervosa diagnoses within the mental health analysis) are comparable across the transmasculine and transfeminine groups in the TGD cohort, while previous literature has established sex differences in presumably cisgender cohorts (0.9% in females and 0.3% in males for anorexia nervosa) [32]. It is essential to note that sex and gender have been conflated and used interchangeably in previous research centering on cisgender cohorts, which necessitates comparisons across gender and sex groups.

Additionally, our data provides context to health arenas that have remained vague in previous research. First, LGBT (lesbian, gay, bisexual, transgender) communities have been reported to have elevated rates of sexually transmitted infections (STIs), but there is less explicit evidence specifically regarding TGD populations as a unique group, particularly including transmasculine individuals [2]. We show a statistically significant increase in viral hepatitis and HIV between the TGD cohort and the control cohort, with both diagnoses more common in the transfeminine cohort specifically. While the prevalence of non-HIV and non-hepatitis STI types (2%) is approximately equivalent between the TGD cohort and the US population [34, 35], it is approximately 50% higher than the control cohort (1%, aPR 1.50, 95% CI(1.21–1.86), p-value = 0.008). Second, LGBT individuals are expected to have worsened digestive health as a result of discrimination and social determinants of health [18], and the TGD cohort has slightly more frequent diagnoses of a personal history of digestive system diseases compared to the control cohort (aPR 1.35, 95% CI (1.18–1.55), p-value <0.001). Conversely, more specific estimates for irritable bowel syndrome and symptoms involving the digestive system were not significant when adjusting for confounders. Third, the absence of significant results is notable in other aspects of the data: there were no gender characteristic differences in the TGD cohort for neurological conditions and sleep-related diagnoses, despite whole population studies demonstrating elevated neurological phenotypes among females compared to males [36,37].

We also document elevated prevalences in the TGD cohort relevant to transition-related care. The frequent diagnosis of "other endocrine disorders" in TGD patients is likely representative of non-specific diagnostic codes utilized to get coverage for gender-affirming hormone therapy (GAHT) [38]. Similarly, the high prevalence of "need for hormone replacement therapy (postmenopausal)", especially in the transfeminine cohort, could also likely be related diagnostic codes leveraged for insurance coverage of GAHT. Other diagnoses, such as secondary erythropoiesis, are expected to be related to transition-related care; similarly, the significant increase of acne in TGD cohorts, with an even higher prevalence in the transmasculine cohort, is an anticipated adverse event of testosterone usage [36].

## Mental health and neurodevelopmental phenotypes

TGD populations are documented to have increased mental and psychiatric comorbidities, theorized to be secondary to increased stressors and structural barriers [39–41]. As anticipated, our TGD dataset has highly elevated rates of mental health conditions, with 70% of the cohort diagnosed with mood disorders. This may reflect a trend in which TGD individuals are more likely to have mood disorders documented as part of qualifying assessments for gender-affirming treatment [42]. Additionally, these findings may include diagnoses that have improved or resolved following gender-affirming care

since we did not conduct a time-series analysis. The prevalence of suicidal ideation or attempt (37%) in our data is notably consistent with previous research on suicidality in TGD populations, and is substantially elevated from non-TGD populations [43–45]. *The observation of heightened suicidality in TGD individuals, given our study's methodology and sample size, highlights the necessity of support and public health intervention for this community.*

*Similarly, t*he significant increase in dissociative disorder in the TGD cohort follows previous research linking transphobia and gender dysphoria to dissociative symptoms, which are known consequences of trauma.(Keating and Muller 2020; Goetz and Arcomano 2023; Dorothy 2025) [46,47] Again, these phenotypes may be alleviated by gender-affirming care that is not captured in our analysis. Interestingly, while there are strongly gendered and/or sex-specific aspects conventionally associated with psychiatric diagnoses, our data showed consistency between gender cohorts with regard to mental health conditions. Previous studies conflict on whether TGD individuals have tobacco, alcohol, and other substance use disorders at a greater rate than cisgender populations [1,2,48]. Our data suggests substantial increases in prevalence compared to the control cohort in all three categories (tobacco use disorder 32% versus 14%; alcohol-related disorders 14% versus 5%; non-alcohol substance use disorders 17% versus 6%) [49]. Our findings did not recapitulate differences in substance use across gender-characteristic cohorts, particularly when adjusting for confounders.

TGD populations have been reported to have higher rates of autism than non-TGD individuals [50], and this is supported by our analysis identifying a prevalence of TGD autistic individuals (7%) twice the rate of the global autism average and three-fold more common than in the control cohort (2%, aPR 3.32, 95% CI (2.99–3.68), p-value < 0.001) [51]. These data are comparable to estimates of TGD autism in smaller studies, including 6% in a study of 532 adults and 7.8% in a study of 204 youths [52]. Transfeminine patients were statistically more likely to have been diagnosed with autism in our cohort (9% versus 6%, aPR 0.48, 95% CI (0.37–0.63), p-value <0.001), which may be secondary to sex differences in autism diagnostic rates prior to coming out/transitioning rather than an actual variation in incidence. ADHD was also elevated in our TGD cohort (21%) compared to the control cohort (9%, aPR 1.61, 95% CI (1.53–1.69), p-value <0.001) and prior literature in the general population (3–7%) [53]. This estimate is comparable to results in one small online volunteer sample of TGD individuals (20.4%) but otherwise elevated compared to previous estimates [52]. Further research is necessary to determine the links between neurodivergence and trans identity and how gender-affirming care can best accommodate and empower patients with autism and ADHD, as well as other mental health conditions shown to be elevated in this community. Gender-affirming care has been shown by previous studies to broadly improve mental health in TGD individuals,(Chen et al. 2023; Cheung et al. 2024; Tordoff et al. 2022) and so future studies may integrate the status and extent of gender-affirming care when evaluating mental health outcomes.

## Broader implications

Our analyses indicate that overarching health disparities across multiple clinical diagnostic categories are present in the TGD population, with limited exceptions. Numerous intersecting factors may contribute to this outcome, including increased interactions with and documentation within healthcare systems in pursuit of gender-affirming care; diagnoses and comorbidities associated with TGD identity (such as autism); and variation in social and structural determinants of health compared to the control cohort. Two explanatory hypotheses have the most aggregate evidence in the literature: 1) the increased "weathering" of TGD individuals and 2) the medicalization of the TGD community.

Weathering describes the early mortality and morbidity present in marginalized communities caused by repeated, socially structured stress akin to the erosive geological process [54]. This phenomenon has primarily been studied in the context of racism affecting Black individuals, but there is accumulating evidence that the health of TGD individuals is also affected by weathering and minority stress [39,55]. In the TGD population, identified social determinants of health that can negatively impact individual wellness include chronic stress, higher rates of homelessness, unemployment, being un- or underinsured, barriers to healthcare, frequent victimization, and elevated Adverse Childhood Events scores [13,56].

The weathering experienced by the TGD community by proximal (or interpersonal) and distal (or structural) stressors likely strongly affects our cohorts' broadly elevated prevalence of clinical phenotypes, although we cannot test for mechanisms within this paper. Our data supports previous literature identifying elevated rates of mental health and neurological conditions in TGD individuals compared to the general population. We additionally note that many of the diagnoses studied here can themselves increase marginalization and augment the weathering and health barriers experienced by TGD individuals. Further, we identified elevated prevalence in the TGD cohort in clinical phenotypes heretofore not described for TGD individuals. Several of these diagnoses have previously been hypothesized to be increased in prevalence in the TGD community based on known factors (for instance, sleep disorders are associated with mental comorbidities [57,58]), and we have provided additional data to endorse further research to meet the specific needs of TGD patients. Additionally, in recognition of our data and others suggesting the increased burden of disability and chronic disease in the TGD population [11,59], it is apparent that structural stigma and experiences of discrimination must decrease in order for the TGD community to have equitable and comparable rates of disease to the cisgender community. In fact, long-term data from the Netherlands on TGD youth indicates that adequate support leads to TGD adults with similar mental health status as their cisgender peers [60]. Given the relationship between mental and physical health, TGD health across categories would likely also be drastically improved by adequate care and support across various social determinants of health. As the impact of political and legislative forces on LGBTQ+ individuals' mental and physical health in the US is a known phenomenon, these trends within TGD communities may be further exacerbated in the coming years.(Schlehofer et al. 2023) While societal and interpersonal pressures on the TGD community may lead to the avoidance of care or lower self-disclosure, such that the anticipated growth of the TGD community is obscured, this does not negate the evolving understanding of TGD patients' unique and urgent healthcare needs.

Nevertheless, additional context to diagnostic rates should be considered when interpreting our data. The medicalization or pathologization of TGD individuals is a known event wherein TGD patients are categorized and treated through clinical language and diagnoses rather than as complex, holistic individuals [1]. This medicalization is most immediately evident by the presence of "gender dysphoria" as a diagnosis in the Diagnostic and Statistical Manual of Mental Disorders, Fifth Edition (DSM-5), but may also contribute to the high prevalence of "other endocrine disorders" as a diagnosis often necessary for insurance coverage of GAHT. Historic racial disparities have been documented in the diagnoses of psychotic disorders based on discrimination against Black patients, and prejudices toward TGD patients leading to pathologization of patient experiences may contribute to the elevation of these diagnoses in our cohort as well [61,62]. Both "schizophrenia and other psychotic disorders" and "psychosis" were significantly elevated in the transfeminine cohort compared to the transmasculine, which may suggest additional marginalization either through biomedical stressors or diagnostic biases. Psychological distress and suicidality have been reported to be higher in transfeminine compared to transmasculine young adults, attributed to structural oppression related to transmisogyny, which may lead to increased weathering in this group.(Wang et al. 2024) Research on manifestations of transmisogyny, or misogyny specific to transfeminine individuals, has discussed the risk of physical and verbal harassment and violence as well as social rejection in cisgender and transgender spaces specific to transfeminine individuals, which may in turn lead to an increase in stress-exacerbated conditions such as psychosis.

Conversely, there is another phenomenon known as "trans broken arm syndrome," which is when there is a misattribution of clinical significance from a patient's chief complaint to their (unrelated) transgender identity [63]. This could be an explanation for the decreased prevalence of IBS and history of digestive disease in our study compared to the general population, as symptoms may have been attributed to GAHT or other factors of TGD identity and/or medical transition [13]. As such, it is essential to evaluate the clinical rigor of diagnoses in the TGD population and how these may lead to over- and under-reporting disease prevalence.

Lastly, it is necessary to be mindful of informative presence bias in EHR data, wherein individuals with known health conditions or needs are likely to have increased EHR data compared to other individuals. There are likely diagnoses

shared in the control group that are not documented, given their fewer interactions with the healthcare system. Controlling for medical visits is often insufficient to account for this bias, given differences in the specificity and sensitivity across medical and mental health diagnoses [64,65]. As such, it may be the case that TGD patients experience overdiagnosis secondary to increased healthcare contact for either conditions related to weathering or in pursuit of gender-affirming care. As this is a prevalence study identifying ICD codes, we do not capture or infer the true incidence of any diagnoses examined.

## Limitations

The identified cohort and its limitations must be considered when evaluating our results, particularly regarding the two gender characteristic groups presented. We believe our cohort captured most individuals who have had access to healthcare related to their TGD status. However, we likely miss others who received care through this system, especially individuals who have not been able to access gender-affirming healthcare due to social, safety, or financial barriers. We additionally inferred gender characteristics rather than using the gold standard of self-reported gender identity, which may lead to misclassification or oversimplification of patient identities. Additionally, the transmasculine group skews much younger in our TGD cohort. There is generally an increased disease burden in our transfeminine cohort that follows previous research on their increased morbidity and mortality, which is unlikely to reflect age variation given adjustment for birth year [66,67]. Additionally, we were unable to separately analyze nonbinary individuals as we only had self-report data to identify them, and the resultant cohort was too small to analyze. As nonbinary individuals are reported to comprise almost a third of TGD individuals, we are likely either missing or miscategorizing a substantial portion, which limits our capability to evaluate their specific health needs [30]. Additionally, the existence of sex-specific diagnostic codes across gender characteristic groups, from postmenopausal diagnoses to male infertility, may reflect how codes are inconsistently or contradictorily applied by providers [38]. Lastly, we report only the birth certificate sex. Despite availability, we elected not to use sex documented from other available vital records or EHR due to the assumed wide variability in practice, policy, and individual choice in completion of those records regarding a reflection of natal sex (assigned sex at birth). We observed mixed sex designations across multiple sources of data, and thus defaulted to a singular source of birth certificate sex for consistency. However, this variable was missing for individuals born out of state. Future studies may benefit from developing reliable methods of triangulation to determine sex assigned at birth for individuals without a birth certificate.

Given that TGD individuals may utilize telehealth, gender-affirming surgeries, or other external avenues of transition-related medical care, there may be TGD individuals we do not capture based on procedural inclusion. We cannot compare outcomes based on GAHT utilization and other gender-affirming procedures. Ideally, sex and gender documentation in electronic healthcare records will improve and lead to greater power to evaluate cohort-specific healthcare needs and protective factors that may result from affirming and competent care [14]. **To this end, we recommend that electronic health records allow individuals to self-identify gender identity, natal sex (assigned sex at birth), and legal gender marker to improve documentation and subsequent studies.** Additionally, further studies can incorporate further context, such as the time of diagnoses and socioeconomic factors, to elucidate outcomes of gender-affirming treatment and aspects specific to subgroups within the TGD cohort.

## Conclusion

We evaluate a range of medical and mental health conditions within a population-ascertained, population-wide cohort of TGD individuals. This research shows that diagnostic rates are elevated in TGD populations across clinical categories outside of the siloes generally considered in TGD health research. These elevated prevalences may reflect the worsened healthcare outcomes of a marginalized population and/or the pathologization of the TGD community reflected in diagnostic processes. Although this study is correlative, the resultant data provides a foundation for causal theoretical frameworks in future research. The adage has been used for research assessing LGBT health disparities by field leaders such as Judith Bradford: "If you're not counted, then you don't count."(Jaffe 2016) [68] We assert that it is essential to critically

evaluate the specific health needs of TGD patients across all aspects of health—physical and mental—so that both the clinical and research bodies involved in TGD care can improve these disparate outcomes.

## Supporting information

**S1 Table. Medical clinical phenotypes.** Prevalence of medical clinical phenotypes for TGD cases and population controls, and association of phenotypes with TGD group membership.
(DOCX)

**S2 Table. Mental health/neurodevelopmental phenotypes.** Prevalence of mental health and neurodevelopmental clinical phenotypes for TGD cases and population controls, and association of phenotypes with TGD group membership.
(DOCX)

**S3 Table. Orthogonal codes.** Diagnostic codes, procedural codes, and medication classes used as orthogonal evidence supporting TGD identity.
(DOCX)

**S4 Table. Phenotypes and PheWAS codes.** Corresponding PheWAS codes for medical and mental health/neurodevelopmental clinical phenotypes of interest.
(DOCX)

**S5 Table. Data subset by gender characteristic.** Demographic characteristics, medical clinical phenotypes, and mental health/neurodevelopmental clinical phenotypes subset by gender characteristic (transmasculine, transfeminine).
(DOCX)

## Acknowledgments

We thank the Pedigree and Population Resource of Huntsman Cancer Institute, University of Utah for its role in the ongoing collection, maintenance, and support of the Utah Population Database (UPDB). We acknowledge the use of UPDB, All Payer Claims Data (APCD), Health Care Facility Data, and University of Utah Health Enterprise Data Warehouse (EDW). The authors would also like to thank UPDB staff members who devoted significant time and effort to data extraction. We thank the University of Utah Pedigree and Population Resource and the University of Utah Health EDW for establishing the Master Subject Index between the Utah Population Database and the University of Utah Health Sciences Center.

## Author contributions

**Conceptualization:** Laurel Hiatt, Amanda V. Bakian, Brooks R. Keeshin, Anna Docherty, Michael Staley, Alison Fraser, Emily Sullivan, Hilary Coon, Anne V. Kirby.

**Data curation:** Laurel Hiatt, Blessing S. Ofori-Atta, Amanda V. Bakian, Nicole L. Mihalopoulos, Anna Docherty, Alison Fraser, Emily Sullivan, Hilary Coon, Anne V. Kirby.

**Formal analysis:** Laurel Hiatt, Blessing S. Ofori-Atta, Amanda V. Bakian, Nicole L. Mihalopoulos, Brooks R. Keeshin, Anna Docherty, Michael Staley, Hilary Coon, Anne V. Kirby.

**Investigation:** Laurel Hiatt, Anna Docherty, Emily Sullivan.

**Methodology:** Laurel Hiatt, Blessing S. Ofori-Atta, Amanda V. Bakian, Nicole L. Mihalopoulos, Brooks R. Keeshin, Anna Docherty, Hilary Coon, Anne V. Kirby.

**Project administration:** Laurel Hiatt, Amanda V. Bakian, Nicole L. Mihalopoulos, Anna Docherty, Michael Staley, Erin A. Kaufman, Hilary Coon, Anne V. Kirby.

**Supervision:** Amanda V. Bakian, Nicole L. Mihalopoulos, Brooks R. Keeshin, Anna Docherty, Michael Staley, Hilary Coon, Anne V. Kirby.

**Validation:** Blessing S. Ofori-Atta, Anna Docherty, Alison Fraser.

**Visualization:** Erin A. Kaufman, Anne V. Kirby.

**Writing – original draft:** Laurel Hiatt, Amanda V. Bakian, Nicole L. Mihalopoulos, Anne V. Kirby.

**Writing – review & editing:** Laurel Hiatt, Amanda V. Bakian, Nicole L. Mihalopoulos, Brooks R. Keeshin, Anna Docherty, Michael Staley, Alison Fraser, Emily Sullivan, Erin A. Kaufman, Hilary Coon, Anne V. Kirby.

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
