## [Decision Letter · Decision Letter 0]

15 Apr 2025

Dear Dr. Hiatt,

We look forward to receiving your revised manuscript.

Kind regards,

Jack Turban MD MHS

Academic Editor

PLOS ONE

2. In the online submission form, you indicated that [Data is available through the Utah Population Database.].

Additional Editor Comments:

Thank you for the opportunity to review this manuscript. It has some great strengths, including sample size, but the reviewers and I had some methodological concerns (especially regarding ascertainment of unique cases). In addition to addressing the reviewers' comments below, please comment on:

- The introduction sets up that past research is based on convenience samples. The data presented also seem to be from a convenience sample. Can you clarify if this is the case?

- While critiquing past studies for being small, it’s important to note that this sample if 6,664 TGD individuals is smaller than other studies (e.g., USTS)

- It’s worth mentioning in the intro that TransPOP exists, as this is truly a probability sample (though ended up being small for this reason)

- It’s important to acknowledge that different datasets have different strengths and weaknesses and that non-probability samples are essential for research on smaller populations (see Turban et al Transgender Health 2023 for a more detailed discussion on this).

- Please comment on the choice to use some unusual diagnostic codes (e.g., F64.1 dual role transvestism — is this consistent with a TGD population?), as reviewer 1 also brought up.

Reviewers' comments:

Reviewer's Responses to Questions

**Comments to the Author**

1. Is the manuscript technically sound, and do the data support the conclusions?

Reviewer #1: Yes

Reviewer #2: Partly

Reviewer #3: Yes

2. Has the statistical analysis been performed appropriately and rigorously?

Reviewer #1: Yes

Reviewer #2: Yes

Reviewer #3: Yes

3. Have the authors made all data underlying the findings in their manuscript fully available?

Reviewer #1: Yes

Reviewer #2: Yes

Reviewer #3: Yes

4. Is the manuscript presented in an intelligible fashion and written in standard English?

Reviewer #1: Yes

Reviewer #2: Yes

Reviewer #3: Yes

Reviewer #1: Thank you for the opportunity to review this interesting and timely manuscript, in which the authors have used 26-year, population-level data from Utah to examine health conditions co-occurring in a gender-diverse population. Their methodology is compelling, and their findings concerning: diverse individuals had higher rates of most conditions examined, particularly in the psychiatric realms. The authors are modest in not over-stating their findings, and direct in sharing their limitations, such as being unable to calculate prevalence rates.

I am enthusiastic about this work and have several inquiries / clarification requests that I hope may further strengthen the manuscript:

1. Please define Phecodes early on (they are mentioned in the abstract). These are useful terms that I (and many readers, I suspect) are not familiar with. The “hierarchical Phecode system” is currently not mentioned until page 12.

2. The umbrella term of gender diverse includes many ICD-10 codes that are obsolete (or even offensive to some). A note of contextualization about “transexual” codes would be welcome, even if as a footnote to the relevant table.

3. On a related note: do all the codes add up to a true umbrella of “transgender and gender diverse (TGD)”? I think so, but am not sure. As TGD seems to be the more commonly used term these days, I would clarify the issue, and noting relevant shortcomings, if any. Stated alternatively: does ICD-10’s “gender incongruence” map neatly onto the generally accepted definition of TGD?

4. Suicide and suicide-related behavior is a major finding, and one with arguably the highest public health burden. I was surprised not to see it addressed in the discussion and think it is imperative to do so. Even if the findings are not necessarily new, they are important replications, especially in light of the study’s methodological rigor and sample size.

5. How does your statement of “an anticipated doubling of the size of the TGD population” resonate with the current political landscape and rhetoric? Although this may be speculative / editorial territory, it could be important to signpost it. For example, will some TGD individuals be more reluctant to seek care, to self-label, etc.? Could the TGD population actually contract over the coming years?

Reviewer #2: The authors use linked administrative data to examine disparities in ICD medical diagnoses between patients classified as transgender vs. non-transgender. The results indicate differences in all outcomes to the detriment of transgender patients, which largely corroborates prior research. The authors should contend with the following comments.

Major

In the introduction, the authors cite self-report as a notable limitation of prior research on transgender health. The bigger issue is the sampling methodology. Self-report is a universal limitation of all survey research, and if one is to elevate it to point of an Achilles heel, then it renders all survey research suspect. The evidence they cite to bolster this point was based more in sampling and response bias than recall bias. The authors should focus their critique on the sampling methodologies.

In the methods, the authors write “It is challenging to give an estimate of how many individuals are within the UPDB, as it contains millions of records derived from various sources.” In the results they write, “Estimating a TGD population prevalence was outside the scope of this study, as the research team did not have access to the relevant denominators…” These statements raise concerns about ascertainment of unique cases (i.e., uniques) in the dataset. Although the UPDB pulls data across multiple systems, it must do so on a unique case basis, which means that a denominator of uniques (i.e., universe) should be known. If uniques cannot be ascertained, then that is a major concern because people’s visits may be counted more than once. The authors need to clearly explain this issue of uniques in the Methods and explain why, if uniques were able to be identified, a denominator could not be ascertained.

The authors explain that “Administrative sex was missing or unknown for 3509

(53%) of TGD individuals and 34925 (54%) of the control cohort, indicating a consistent lack of documentation across cohorts.” Assumed consistency notwithstanding, and although this may be more understandable for TGD people for whom administrative data is fraught, it is less clear why more than half of assumedly cisgender people were missing administrative sex data. The authors should explain this lack of very basic data in EHR or why sex wasn’t available from vital records.

The authors control for the number of ICD codes as a way of getting at informed presence bias, but the number of ICD codes does not account for disease severity and comorbidity. Because they have access to ICD diagnosis data, the authors may want to use a weighted comorbidity index score (e.g., weighted Elixhauser or Charlson), which is better at accounting for the severity of diagnosed conditions than a simple count.

The authors write: “Both “schizophrenia and other psychotic disorders” and “psychosis” were significantly elevated in the transfeminine cohort compared to the transmasculine, which may suggest additional marginalization either through biomedical stressors or diagnostic biases.” This sentence needs more explanation because it is unclear what kind of biomedical stressors or diagnostic biases that would specifically target transfeminine individuals and not transmasculine individuals or cisgender women.

Minor

The attribution of the quote “If you’re not counted, then you don’t count” should be attributed to Judy Bradford’s research legacy, which predates the citation given. See: Jaffe, S. (2016). Judith Bradford: a pioneer of research on LGBT health. The Lancet, 387(10023), 1048.

An additional limitation is potential misclassification because gender identity (transmasculine and transfeminine) were assumed by the research team based on other data in the patients’ records. This is triangulation, but it is not the gold standard of self-reported gender identity.

Reviewer #3: Thank you so much for the opportunity to review your article “Unveiling Health Disparities: Diagnostic Prevalences in a Transgender Cohort versus Matched Controls.” This is a great addition to the literature, and offers both an overview of co-occurring medical and mental health diagnoses in TGD individuals in Utah and offers some astute hypotheses about why this may be the case.

In order to make this paper stronger, I offer the following recommendations:

1. In your statement on page 12 it says, “It is challenging to give an estimate of how many individuals are within the UPDB, as it contains millions of records derived from various sources.” Can you give examples or further define “various sources”?

2. Please write out PheWAS acronym the first time it is introduced (page 13).

3. At the beginning of the discussion, you state “as this analysis would be challenging or impossible in most U.S. states.” Can you clarify? Is this due to the current hostility toward medical care for TGD folks? If so, this should be noted in the discussion.

4. On page 18, you state “The significant increase in dissociative disorder in the TGD cohort follows previous research linking transphobia and gender dysphoria to dissociative symptoms.” It may also be important to directly name this as trauma sxs/stressors (e.g. CPTSD).

5. Further down on page 18, you state that “Further research is necessary to determine the links between neurodivergence and trans identity and how gender-affirming care can best accommodate and empower patients with autism and ADHD, as well as other mental health conditions shown to be elevated in this community.” There have been multiple studies on the impact of GAC on MH of patients. It would be helpful to note some of these here (e.g. Chen et al, 2023; Dutton et al 2024; Tordoff, 2022).

6. On page 21 you note, “Ideally, future studies will improve as sex and gender documentation are improved, and there will be greater power to evaluate cohort-specific healthcare needs and the protective factors that may result from affirming and competent care.” I wonder if it would be helpful to separate this idea out as a recommendation (along with the timeline of dx to see how GAC may impact dx), and to briefly discuss the potential challenges to this in the current political/legislative climate.

**Do you want your identity to be public for this peer review?** For information about this choice, including consent withdrawal, please see our Privacy Policy

Reviewer #1: **Yes: ** Andres Martin, MD, PhD

Reviewer #2: No

Reviewer #3: No

---

## [Author Response · Author response to Decision Letter 1]

29 May 2025

We have responded to all reviewer and editor comments in the attached document "Response to Reviewers". Thank you all again for your helpful feedback and guidance!

---

## [Decision Letter · Decision Letter 1]

17 Jul 2025

Dear Dr. Hiatt,

Thank you for submitting your manuscript to PLOS ONE. After careful consideration, we feel that it has merit but does not fully meet PLOS ONE’s publication criteria as it currently stands. Therefore, we invite you to submit a revised version of the manuscript that addresses the points raised during the review process.

We look forward to receiving your revised manuscript.

Kind regards,

Jack Turban

Academic Editor

PLOS ONE

Journal Requirements:

Reviewers' comments:

Reviewer's Responses to Questions

**Comments to the Author**

Reviewer #1: All comments have been addressed

Reviewer #2: (No Response)

2. Is the manuscript technically sound, and do the data support the conclusions?

Reviewer #1: Yes

Reviewer #2: Yes

3. Has the statistical analysis been performed appropriately and rigorously?

Reviewer #1: No

Reviewer #2: Yes

4. Have the authors made all data underlying the findings in their manuscript fully available?

Reviewer #1: Yes

Reviewer #2: No

5. Is the manuscript presented in an intelligible fashion and written in standard English?

Reviewer #1: Yes

Reviewer #2: Yes

Reviewer #1: Thank you for your thoughtful response to my critiques. Thank you for your thoughtful response to my critiques.

Reviewer #2: The authors responded to the reviewers' comments in thoughtful, respectful ways. One minor suggestion is that the issue of the missing sex variable should be in the limitations. The authors explain that the missingness was due to the choice of relying on birth certificate sex from the database. This clearly was a fraught choice (especially if there were other forms of sex data that could have been gathered). The authors can very briefly mention that the missing sex data was a limitation and that future research should gather multiple sex-related data in administrative records that would be used for triangulating patients' sex. This would be an important "lesson learned" for future researchers who may embark on a similar endeavor.

**Do you want your identity to be public for this peer review?** For information about this choice, including consent withdrawal, please see our Privacy Policy

Reviewer #1: **Yes: ** Andrés Martin, MD, PhD

Reviewer #2: No

---

## [Author Response · Author response to Decision Letter 2]

18 Jul 2025

In response to the remaining suggestion by Reviewer 2:

“One minor suggestion is that the issue of the missing sex variable should be in the limitations. The authors explain that the missingness was due to the choice of relying on birth certificate sex from the database. This clearly was a fraught choice (especially if there were other forms of sex data that could have been gathered). The authors can very briefly mention that the missing sex data was a limitation and that future research should gather multiple sex-related data in administrative records that would be used for triangulating patients' sex. This would be an important "lesson learned" for future researchers who may embark on a similar endeavor.”

We have added the following section to the first paragraph of our limitations section:

“Lastly, we report only the birth certificate sex. Despite availability, we elected not to use sex documented from other available vital records or EHR due to the assumed wide variability in practice, policy, and individual choice in completion of those records regarding a reflection of natal sex (assigned sex at birth). We observed mixed sex designations across multiple sources of data, and thus defaulted to a singular source of birth certificate sex for consistency. However, this variable was missing for individuals born out of state. Future studies may benefit from developing reliable methods of triangulation to determine sex assigned at birth for individuals without a birth certificate.”

Thank you again for this opportunity to share our research.

---

## [Editor Report · Decision Letter 2]

23 Jul 2025

Unveiling health disparities: Diagnostic prevalences in a transgender cohort versus matched controls

PONE-D-25-11456R2

Dear Dr. Hiatt,

We’re pleased to inform you that your manuscript has been judged scientifically suitable for publication and will be formally accepted for publication once it meets all outstanding technical requirements.

Kind regards,

Jack Turban

Academic Editor

PLOS ONE

---

## [Editor Report · Acceptance letter]

PONE-D-25-11456R2

PLOS ONE

Dear Dr. Hiatt,

I'm pleased to inform you that your manuscript has been deemed suitable for publication in PLOS ONE. Congratulations! Your manuscript is now being handed over to our production team.

Kind regards,

on behalf of

Dr. Jack Turban

Academic Editor

PLOS ONE